# Acyclic Triterpenoid Isolated from *Alpinia katsumadai* Alleviates Formalin-Induced Chronic Mouse Paw Inflammation by Inhibiting the Phosphorylation of ERK and NF-κB

**DOI:** 10.3390/molecules25153345

**Published:** 2020-07-23

**Authors:** Hyung Jin Lim, Seon Gyeong Bak, Hee Ju Lim, Seung Woong Lee, Soyoung Lee, Sae-Kwang Ku, Sang-Ik Park, Seung-Jae Lee, Mun-Chual Rho

**Affiliations:** 1Immunoregulatory Material Research Center, Korea Research Institute of Bioscience and Biotechnology, Jeongeup-si, Jeonbuk 56212, Korea; lhjin@kribb.re.kr (H.J.L.); tsk9096@kribb.re.kr (S.G.B.); limhj09@kribb.re.kr (H.J.L.); lswdoc@kribb.re.kr (S.W.L.); sylee@kribb.re.kr (S.L.); 2Department of Bioactive Material Sciences, Chonbuk National University, Jeonju-si, Jeonbuk 54896, Korea; 3Department of Marine Bio Food Science, Chonnam National University, Korea, Yeosu-si, Jeonnam 59626, Korea; 4Division of Biotechnology and Advanced Institute of Environment and Bioscience, Jeonbuk National University, Iksan-si, Jeonbuk 54596, Korea; 5Department of Anatomy and Histology, College of Korean Medicine, Daegu Haany University, Gyeongsan-si, Gyeonbuk 38610, Korea; gucci200@hanmail.net; 6College of Veterinary Medicine, Chonnam National University, Gwangju-si 61186, Korea; sipark@jnu.ac.kr

**Keywords:** *Alpinia katsumadai*, 2,3,5,22,23-pentahydroxy-2,6,10,15,19,23-hexamethyl-tetracosa-6,10,14,18-tetraene, chronic mouse model, anti-inflammation

## Abstract

Chronic and excessive inflammation can destroy host organs and cause inflammatory diseases such as inflammatory bowel disease, asthma, and rheumatoid arthritis. In this study, we investigated the anti-inflammatory effects of *Alpinia katsumadai* seed-derived 2,3,5,22,23-pentahydroxy-2,6,10,15,19,23-hexamethyl-tetracosa-6,10,14,18-tetraene (PHT) using lipopolysaccharide (LPS)-stimulated J774 cells and a formalin-induced chronic paw inflammation mouse model. The in vitro results showed that PHT exhibited no cytotoxicity and decreased LPS-induced NO secretion. Additionally, PHT inhibited LPS-induced inducible NO synthase (iNOS) and cyclooxygenase 2 (COX2) protein expression. The quantitative real-time PCR results showed that PHT downregulated the gene expression of the proinflammatory cytokines interleukin-1β (IL-1β) and interleukin-6 (IL-6) but not tumor necrosis factor α (TNF-α). PHT inhibited the LPS-induced phosphorylation of extracellular signal-regulated kinase (ERK) and nuclear factor kappa light chain enhancer of activated B cells (NF-κB). In a mouse model, oral administration of 50 mg/kg PHT significantly alleviated both mouse paw thickness and volume. These results indicate that PHT has potential anti-inflammatory effects and should be considered a possible functional material.

## 1. Introduction

The inflammatory response is a defense mechanism of the body against pathogenic factors such as bacterial and viral infection, irradiating substances and damaged cells [1]. Detection of these factors by pattern recognition receptors (PRRs) initiates the inflammatory response [2]. Toll-like receptors (TLRs) are PRRs that respond to various agonists in a subtype-dependent manner. Lipopolysaccharide (LPS) binds with LPS binding protein (LBP), and this complex activates the TLR4 receptor. Activated TLR4 leads to the phosphorylation of c-Jun *N*-terminal kinases (JNK), extracellular signal-regulated kinase (ERK), p38 mitogen activated protein kinase (MAPK) and nuclear factor kappa light chain enhancer of activated B cells (NF-κB) [1,3]. Phosphorylated MAPKs activate the transcription factor activator protein-1 (AP-1). AP-1 and NF-κB act as transcription factors in the nucleus and upregulate pro-inflammatory genes [3,4,5]. Upregulated pro-inflammatory cytokines and chemokines mediate and maintain the inflammatory response and stimulate immune cells to remove pathogenic factors [1,6]. However, excessive production of inflammatory cytokines and chemokines caused by chronic and acute inflammatory conditions could result in self-destructive reactions and contribute to inflammatory diseases such as inflammatory bowel disease, asthma, and rheumatoid arthritis [7,8,9,10,11].

*Alpinia katsumadai* Hayata (*Alpinia katsumadae* Hayata) is a synonym of the *Alpinia hainanensis* K. Schum. and belongs to the family *Zingiberaceae*. Its seed has been used as a traditional Chinese medicine to treat inflammatory and digestive diseases, and various biological properties of extracts of this plant have been reported, such as anti-emetic, anti-proliferative, anti-viral and anti-asthmatic effects [12,13,14,15]. It is known that the phytochemical constituents of *A. katsumadai* are diarylheptanoids, monoterpenes, sesquiterpenoids, flavonoids and chalcones [16]. Acyclic triterpenoid, which is a subtype of triterpenoid, is found in various natural materials such as algae, bacteria and plants [17,18,19]. Its biological activities have been reported as anti-inflammatory, anti-oxidative, anti-plasmodial- and cholesterol-lowering [19,20,21,22]. In a previous study, we isolated four acyclic triterpenoids from *A. katsumadai* seeds and reported inhibitory effects on interleukin-6 (IL-6)-induced signal transducer and transcription 3 (STAT3) phosphorylation, which is a major mediator of inflammation and cancer [16]. Among the tested substances, 2,3,5,22,23-pentahydroxy-2,6,10,15,19,23-hexamethyl-tetracosa-6,10,14,18-tetraene (PHT) and 2,3,22,23-tertrahydroxy-2,6,10,15,19,23-hexamethyl-tetracosa-6,10,14,18-tetraene inhibited the binding of intercellular adhesion molecule 1 (ICAM1) to THP-1 monocytes [22].

In this study, we investigated the anti-inflammatory effects of PHT, an acyclic triterpenoid isolated from *A. katsumadai* seeds, on LPS-induced J774 cells and its therapeutic effects against chronic inflammatory disease in a formalin-induced mouse paw inflammation model.

## 2. Results and Discussion

### 2.1. PHT Decreases NO and PGE2 Production in LPS-Induced J774 Cells

Under inflammatory conditions, high levels of NO and PGE2 were observed [23,24]. These factors are major inflammatory mediators and are implicated in the pathogenesis of inflammatory disorders. Thus, inhibition of NO and PGE2 could alleviate inflammatory diseases [25,26]. Prior to investigating the anti-inflammatory effect of PHT (Figure 1A), the cytotoxicity of PHT was measured by the MTT assay. There was no PHT-mediated cytotoxicity in J774 cells (Figure 1B). Next, we performed a NO assay and PGE2 ELISA on LPS-induced J774 murine macrophages to evaluate the anti-inflammatory effect of PHT and found that PHT decreased both NO and PGE2 levels in a dose-dependent manner (Figure 1C,D).

### 2.2. PHT Suppresses Inducible NO Synthase (iNOS) and Cyclooxygenase 2 (COX2) Expression in LPS-Induced J774 Cells

NO and PGE2 are synthesized by NOSs and COXs. There are several isotypes of these enzymes. Three distinct NOS isoforms exist: endothelial NOS (eNOS), neural NOS (bNOS) and iNOS. eNOS and bNOS are constitutively expressed and regulated by intracellular Ca^2+^ concentrations [25,27]. Excessive NO production during the inflammatory reaction is related to iNOS, which is regulated by cytokines. COXs have two isoforms: COX1 and COX2. COX1 is constitutively expressed in most cells. COX2 is expressed at low levels in most cells under normal conditions but is dramatically upregulated in response to inflammatory stimuli in immune cells [26,28]. To assess the effect of PHT on LPS-induced iNOS and COX2 protein expression, an immunoblot assay was performed. LPS-induced iNOS expression was significantly inhibited by 10 μM PHT treatment, and LPS-induced COX2 expression was significantly inhibited by 3 and 10 μM PHT treatment (Figure 1E–G).

### 2.3. PHT Downregulates IL-1β and IL-6 But Not TNF-α Gene Expression in LPS-Induced J774 Cells

Next, we investigated the effect of PHT on LPS-induced pro-inflammatory gene expression. Pro-inflammatory cytokines are used as inflammatory markers, and excessive inflammatory cytokine levels, called cytokine storms, can cause organ failure [29,30,31]. Treatment with PHT significantly downregulated the gene expression of IL-1β and IL-6, but not TNF-α (Figure 2). However, PHT treatment induced a decreasing trend in TNF-α gene expression in a dose-dependent manner (Figure 2C).

### 2.4. PHT Inhibits the Phosphorylation of ERK and NF-κB p65 But Not JNK or p38 in LPS-Induced J774 Cells

In general, LPS binds to its receptor TLR4 and activates downstream signaling pathways, including MAPK and NF-κB. Immunoblot analysis was performed to determine whether PHT affects the MAPK and NF-κB signaling pathways. As shown, PHT inhibited the phosphorylation of ERK and NF-κB p65 but not JNK or p38 (Figure 3). Some previous studies reported that there were no significant decreases in LPS-induced TNF-α levels or p38 phosphorylation levels, but other pro-inflammatory cytokines and NF-κB phosphorylation were decreased by PHT treatment [32,33,34]. Previous study reported that TNF-α expression is regulated in distinct MAPK pathways in different macrophage populations [35]. In microglia cells, it was reported that selective inhibition of phosphorylated p38 or JNK decreased TNF-α expression, but inhibition of phosphorylated NF-κB alone did not affect TNF-α expression [36]. Taken together with our data, it seems that TNF-α expression in J774 cells was maintained by p38 or JNK MAPK regulated signaling such as LPS- induced TNF-α factor (LITAF), cAMP response element binding protein (CREB) and activating transcription factor 2 (ATF2) [35,36,37].

### 2.5. PHT Alleviates Formalin-Induced Chronic Mouse Paw Inflammation

Subsequently, we assessed the anti-inflammatory effect of PHT on a formalin-induced chronic paw inflammation mouse model. The formalin-induced paw inflammation model is widely used to evaluate the anti-inflammatory effects of substances [38,39]. There were no significant differences in body weights or intact paw thickness in the PHT group compared with the formalin control group (Figure 4A,B). However, the body weights of the dexamethasone-injected group were significantly decreased beginning on the third day after administration (Figure 4A). There was a decreasing trend in the intact paw thickness of the dexamethasone-treated group, but this trend was not significant (Figure 4B). The induced paw thicknesses were significantly decreased in both the dexamethasone- and PHT-treated groups beginning on the first and second days after administration, respectively (Figure 4C). The differences in paw thicknesses in the dexamethasone- and PHT-treated groups were significantly decreased beginning on the second day after administration (Figure 4D). Finally, there was a significant decrease in the induced paw volumes in the dexamethasone- and PHT-treated groups beginning on the first and second days after administration, respectively.

Histological analysis of the induced paw dorsum pedis and digital skin of mice in the formalin control group showed severe fibrosis and infiltration of immune cells, which led to hypertrophy of the subcutaneous regions (Figure 5A,B). In particular, a decrease in bone mass, which is a sign of osteoarthritis, was observed in the formalin control group (Figure 5B). However, dexamethasone and PHT treatment dramatically improved these effects (Figure 5A,B). These results show that oral administration of PHT effectively mitigates formalin-induced paw inflammation without changes in body weight.

## 3. Materials and Methods

### 3.1. Materials and Reagents

The mouse macrophage cell line J774 was obtained from the American Type Culture Collection (ATCC; Rockville, MD, USA) and maintained with DMEM (GibcoBRL; Grand Island, NY, USA) supplemented with 10% FBS, 50 U/mL penicillin and 50 mg/mL streptomycin at 37 °C in a 5% CO_2_ incubator. Thiazolyl blue tetrazolium bromide for the MTT assay, Griess reagent for the NO assay and dexamethasone were purchased from Sigma Aldrich (St. Louis, MO, USA). All antibodies for western blot analysis were obtained from Cell Signaling Technology (Danvers, MA, USA).

### 3.2. Isolation of PHT

#### 3.2.1. General Experimental Procedure

*A. katsumadai* extract was fractionated using silica gel (Kieselgel 60, 230−400 mesh, Merck, Darmstadt, Germany) and RP-C18 silica gel (YMC*GEL ODS-A, 12 nm S-150 µm YMC Co. LTD, Tokyo, Japan) column chromatography (CC). The semi-preparative HPLC system consisted of a HITACHI L-2130 pump (HITACHI, Tokyo, Japan) equipped with a HITACHI UV detector L-2400 using YMC-Pack ODS-H80 column (5 mm, 250 × 20 mm). The structure of PHT was analyzed by 1H-NMR (500 MHz), 13C-NMR (125 MHz), HMQC, and HMBC spectra obtained on a Bruker Biospin Avance 500 spectrometer (Bruker, Billerica, MA, USA) with CDCl3 as a solvent. ESI-MS was conducted using a Shimadzu LCMS-IT-TOF mass spectrometer (Shimadzu, Tokyo, Japan). Optical rotations were determined on a JASCO DIP-370 polarimeter (JASCO, Easton, MD, USA).

#### 3.2.2. Extraction and Isolation of PHT

The detailed separation and purification process of PHT was previously reported [16] and is briefly described in this paper. The seeds of *A. katsumadai* (1.8 kg) were purchased from a herbal market (Daejeon, Korea). The authenticity of the plants was confirmed by Prof. Y. H. Kim, at the College of Pharmacy of Chungnam National University, Daejeon, Korea. A voucher specimen (PBC-386A) was deposited in the Korea Plant Extract Bank at the Korea Research Institute of Bioscience and Biotechnology. The dried seeds were extracted with EtOH (10 L) for 7 days at room temperature and the EtOH solution was concentrated under vacuum to yield a residue (180 g). The residue was suspended in H2O (3 L), and the aqueous layer was partitioned with CHCl3 (10 L). The CHCl3 layer (85g) was subjected to silica gel (150 g) column chromatography using a gradient solvent system of CHCl3-CH3OH (100:0, 90:1, 70:1, 50:1, 30:1, 15:1, 5:1, and 1:1; each 3 L, *v*/*v*) to obtain 22 fractions (F1~22). Fraction F9 (700 mg) was subjected to RP-C18 silica gel (YMC*GEL ODS-A, 100 g) column chromatography eluted with CH3OH-H2O (50:1, 60:1, 70:1, 80:1, 90:1, and 100:0; each 2 L, *v*/*v*) to yield seven sub-fractions (F9-1~7). Fraction F9-3 (300 mg) was subjected to semi-preparative HPLC (YMC-Pack ODS-H80 column, 5 μm, 250 × 20 mm, flow rate 6 mL/min) using isocratic elution 70% CH3CN in H2O to afford compounds PHT (100 mg, tR 30 min).

#### 3.2.3. 2,3,5,22,23-Pentahydroxy-2,6,10,15,19,23-Hexamethyl-Tetracosa-6,10,14,18-Tetraene (PHT)

Yellow oil; C_30_H_54_O_5_; [*α*]$\begin{array}{c} 20\\ D \end{array}$ + 4.8 (c 1.0, CHCl_3_); IR (neat) *ν*_max_ 3400, 2900 cm^−1^; UV(MeOH) λ_max_ (logε) nm: 210 (2.51); HRESI-MS: *m/z* 493.3897 [M-H]^+^ (calcd for C_30_H_53_O_5_, 493.3893); ^1^H-NMR (500 MHz, CDCl_3_) δ_H_ 5.43 (1H, t, *J* = 6.4 Hz, H-7), 5.19 (1H, t, *J* = 6.4 Hz, H-18), 5.14 (2H, m, H-11, H-14), 4.26 (1H, dd, *J* = 7.6, 4.8 Hz, H-5), 3.62 (1H, dd, *J* = 3.6, 8.4 Hz, H-3), 3.35 (1H, d, *J* = 10.4 Hz, H-22), 2.23 (2H, m, H_2_-20), 2.13 (2H, m, H_2_-8), 2.10 (2H, m, H_2_-17), 2.09 (2H, m, H_2_-21), 2.05 (2H, m, H_2_-9), 2.02 (2H, m, H_2_-16), 1.63 (2H, m, H_2_-4), 1.63 (3H, s, H_3_-26), 1.62 (3H, s, H_3_-27), 1.61 (3H, s, H_3_-28), 1.60 (3H, s, H_3_-29), 1.59 (2H, m, H_2_-13), 1.41 (2H, m, H_2_-12), 1.20 (3H, s, H_3_-1), 1.19 (3H, s, H_3_-24), 1.17 (3H, s, H_3_-25), 1.15 (3H, s, H_3_-30): ^13^C-NMR (125 MHz, CDCl_3_) δ_C_ 11.9 (C-26), 16.1 (C-27), 16.2 (C-28, C-29), 23.5 (C-25), 24.0 (C-1), 26.2 (C-21), 26.4 (C-25), 26.6 (C-8, C-17, C-30), 28.4 (C-16), 29.8 (C-12), 36.1 (C-4), 37.0 (C-20), 39.4 (C-9), 39.8 (C-13), 72.8 (C-23), 73.3 (C-2), 78.5 (C-3), 78.6 (C-5), 78.9 (C-22), 124.7 (C-11), 124.9 (C-14), 125.3 (C-18), 126.6 (C-7), 134.9 (C-6), 135.1 (C-10), 135.2 (C-15), 137.3 (C-19).

### 3.3. NO Assay and MTT Assay

J774 mouse macrophages were seeded in 96-well plates at a density of 1 × 10^5^ cells/well for the NO assay and at 3 × 10^4^ cells/well for the MTT assay. For the NO assay, the cells were pretreated with 10 μM dexamethasone or 1, 3, or 10 μM PHT for 1 h before treatment with 200 ng/mL LPS for 18 h. Then, 50 μL of Griess reagent and 50 μL of supernatant were added to a 96-well plate and incubated for 15 min. After incubation, the absorbance was measured at 540 nm using a microplate ELISA reader (Molecular Devices, Sunnyvale, CA, USA). For the MTT assay, the cells were treated with the indicated concentrations of dexamethasone and PHT for 24 h. After incubation, the cells were treated with 10 μL of thiazolyl blue tetrazolium bromide (MTT) solution (5 mg/mL in PBS) for 3 h, and the supernatant was discarded. The remaining formazan crystals were suspended in dimethyl sulfoxide (DMSO), and the absorbance was measured at 540 nm using a microplate ELISA reader.

### 3.4. ELISA

J774 mouse macrophages were seeded in 12-well plates and pretreated with 10 μM dexamethasone or 1, 3 or 10 μM PHT for 1 h before treatment with 200 ng/mL LPS for 18 h. After incubation, the supernatant was collected, and the prostaglandin E2 (PGE2) concentration was measured by a mouse PGE2 ELISA kit (R&D Systems, Minneapolis, MN, USA) according to the manufacturer’s instructions. The absorbance was measured at 450 nm using a microplate ELISA reader.

### 3.5. Quantitative Real-Time PCR

J774 cells were pretreated with 10 μM dexamethasone or 1, 3 or 10 μM PHT and were then treated with 200 ng/mL LPS for 12 h. Whole cells were collected, and total RNA was extracted using the PureLink RNA mini kit (Invitrogen, San Diego, CA, USA) according to the manufacturer’s instructions. The RNA concentration was measured by a microspectrophotometer (Allsheng, Hangzhou, Zhejiang, China), and complementary DNA (cDNA) synthesis was performed. cDNA was synthesized using a PrimeScript 1st strand cDNA synthesis kit (Takara Bio Inc., Shiga, Japan) from 1 μg/mL total RNA. Real-time PCR was performed by a StepOnePlus Real-Time PCR System using TaqMan probes and TaqMan Real-Time PCR master mix (Applied Biosystems, Foster City, CA, USA). The real-time PCR results were normalized to the mouse GAPDH gene.

### 3.6. Immunoblot Analysis

J774 cells were seeded in 6-well plates and pretreated with dexamethasone and PHT for 1 h. After pretreatment, the cells were stimulated with 200 ng/mL LPS for the indicated times. Whole cells were collected, and total proteins were extracted using cell lysis buffer (Cell Signaling Technology). The total protein concentration was measured by a DC protein assay kit (Bio-Rad, Contra Costa County, CA, USA). Equal amounts of protein were loaded and separated on a 4–12% SDS-PAGE gel. The separated proteins were transferred onto a polyvinylidene fluoride (PVDF) membrane and blocked with tris-buffered saline (TBS) containing 5% bovine serum albumin (BSA). Next, the membrane was incubated overnight with primary antibodies at 4 °C. Then, the membrane was incubated with horseradish peroxide (HRP)-conjugated secondary antibodies for 1 h at room temperature and developed using a West-Queen RTS western blot detection kit (iNtRON Bio., Seongnam, Korea).

### 3.7. Animals and Induction of Formalin-Induced Chronic Mouse Paw Inflammation Model

Six-week-old male ICR mice were purchased from OrientBio (Seongnam, Korea). The mice were randomly divided into four groups (*n* = 9 mice per group): intraperitoneal and subaponeurotic injection of saline in mice (intact control), intraperitoneal and subaponeurotic injection of saline and formalin, respectively, in mice (formalin control), intraperitoneal and subaponeurotic injection of dexamethasone and formalin, respectively in mice, and oral and subaponeurotic administration of PHT and formalin, respectively, in mice. After stabilization for one week, the mice were administered saline and 15 mg/kg dexamethasone intraperitoneally and 50 mg/kg PHT orally once a day for 10 days. Chronic inflammation of the paw was induced by subaponeurotic injection of 0.02 mL of 3.75% formalin into the left hind paw 1 h before treatment administration on the first and third experimental days. Body weight, paw thickness, and paw length (long and short axes) were measured daily. At the end of the experiment, the mice were sacrificed, and the mouse paws were removed and fixed in 10% formalin. For histological analysis, the fixed mouse paws were embedded in paraffin, sectioned and stained with hematoxylin and eosin (H&E) according to the general procedure. The experimental protocols were approved by the Institutional Animal Care and Use Committee of Korea Research Institute of Bioscience and Biotechnology (permit number: KRIBB-AEC-19184). All mice were treated according to the Guide for the Care and Use of Laboratory Animals published by the US National Institutes of Health.

### 3.8. Statistical Analysis

The in vitro results are presented as the mean ± standard deviation (SD) of three individual experiments. Statistical analysis was performed using Prism 5 software (GraphPad Software, San Diego, CA, USA). The in vitro data are presented as the mean ± SD of nine individual experiments. Statistical significance was determined by one-way ANOVA followed by Tukey’s test for multiple comparisons.

## 4. Conclusions

In this study, we evaluated cytotoxicity and the inhibition of NO and PGE2 production. Inflammatory cytokine gene expression was analyzed by quantitative real-time PCR. Immunoblot analysis of MAPK and NF-κB signaling molecules was performed to evaluate the effect of *A. katsumadai*-derived acyclic triterpenoid, PHT on LPS-induced signaling in J774 macrophages. It was found that PHT is specifically involved in inhibiting the LPS-induced phosphorylation of ERK and p65, which are involved in various inflammation-related signaling pathways. However, PHT was not involved in JNK or p38 signaling. Taken together with previous study about the inhibitory effect on IL-6/STAT3 which is a major inflammatory signal, our present data confirmed the potent ability of PHT to inhibit inflammation in vitro and in vivo. In conclusion, this triterpenoid may be a useful candidate as an anti-inflammatory agent.

## Figures and Tables

**Figure 1 molecules-25-03345-f001:**
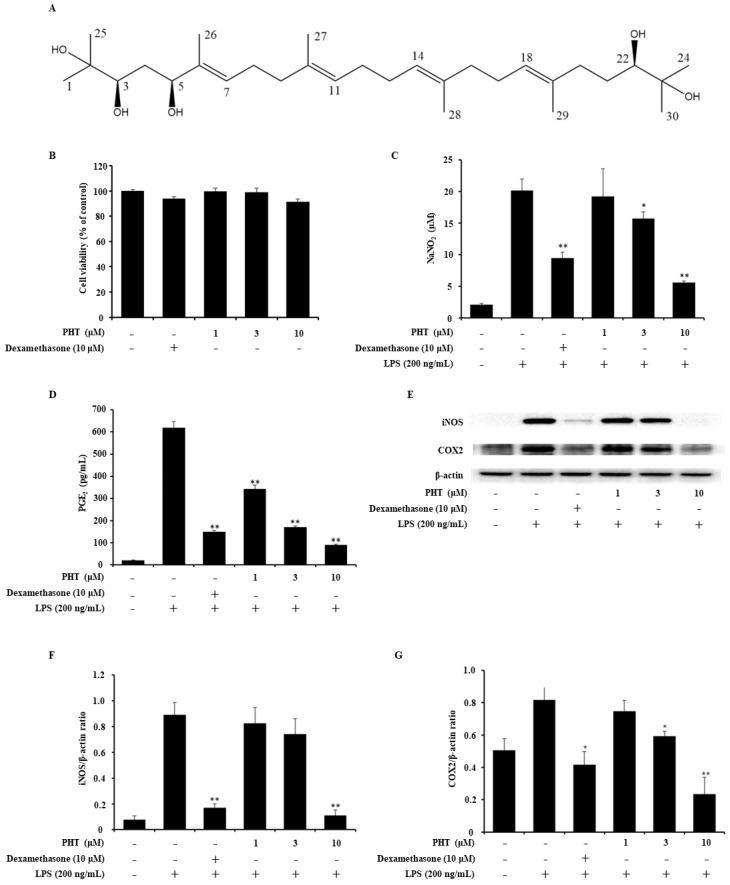
Effects of PHT on LPS-induced NO and PGE2 secretion and iNOS and COX2 protein expression in J774 cells. (**A**) Chemical structure of PHT. (**B**) Cytotoxicity of PHT. J774 cells were treated with 10 μM dexamethasone or 1, 3 or 10 μM PHT for 24 h. Cytotoxicity was determined by the MTT assay. (**C**,**D**) Inhibition of NO and PGE2 production by PHT. J774 cells were treated with LPS (200 ng/mL) for 18 h after pretreatment with 10 μM dexamethasone or 1, 3, or 10 μM PHT for 1 h. (**C**) The concentration of NO was measured by the NO assay using Griess reagents. (**D**) The level of secreted PGE2 was determined by ELISA. (**E**–**G**) PHT treatment decreased iNOS and COX2 protein expression. J774 cells were pretreated with dexamethasone and PHT for 1 h before LPS (200 ng/mL) treatment for 18 h. (**E**) Protein expression levels of iNOS and COX2 were measured by immunoblot assays. The band optical densities of (**F**) iNOS and (**G**) COX2 were calculated by ImageJ software. The immunoblot data are representative of three independent experiments. The values are presented as the mean ± SD of three individual experiments. * *p* < 0.05, ** *p* < 0.01 compared with the LPS-alone group.

**Figure 2 molecules-25-03345-f002:**
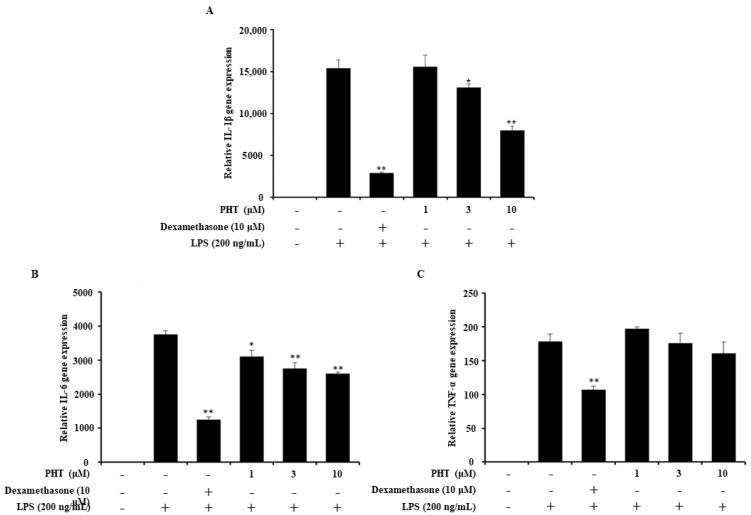
Effect of PHT on LPS-induced proinflammatory gene expression. J774 cells were seeded and pretreated with 10 μM dexamethasone or 1, 3, or 10 μM PHT for 1 h before LPS (200 ng/mL) treatment for 12 h. Total RNA was extracted and used to synthesize cDNA. (**A**) IL-1β, (**B**) IL-6 and (**C**) TNF-α gene expression were measured by quantitative real-time PCR. The gene expression level was normalized to GAPDH expression and is presented as the fold change relative to the untreated group. The values are presented as the mean ± SD of three individual experiments. * *p* < 0.05, ** *p* < 0.01 compared with the LPS-alone group.

**Figure 3 molecules-25-03345-f003:**
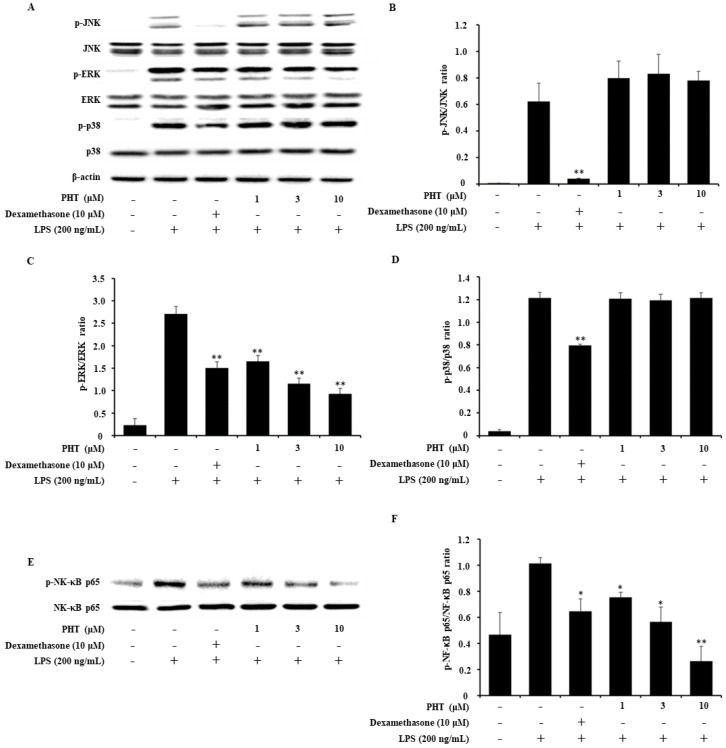
Effect of PHT on the LPS-induced MAPK and NF-κB signaling pathways. J774 cells were treated with LPS (200 ng/mL) for 30 min to 1 h after pretreatment with 10 μM dexamethasone or 1, 3 or 10 μM PHT for 1 h. (**A**) MAPK expression was analyzed by immunoblotting. The band optical densities of (**B**) JNK, (**C**) ERK and (**D**) p38 were measured. (**E**) NF-κB p65 subunit expression was measured by immunoblot analysis, and (**F**) the band optical density was calculated. The immunoblot data are representative of three independent experiments. The values are presented as the mean ± SD of three individual experiments. * *p* < 0.05, ** *p* < 0.01 compared with the LPS-alone group. The optical densities were analyzed by ImageJ software.

**Figure 4 molecules-25-03345-f004:**
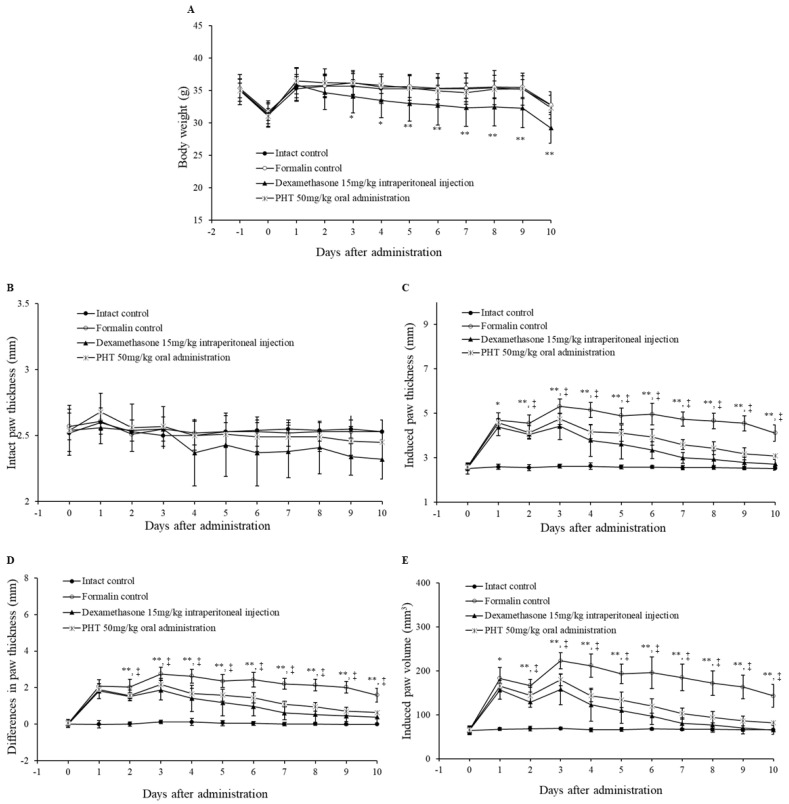
Changes in body weight, intact and induced paw thickness and induced paw volume. Six-week-old male ICR mice were intraperitoneally injected with saline or dexamethasone (15 mg/kg) or orally administered PHT (50 mg/kg) once a day for 10 days. Chronic inflammation was induced by subaponeurotic injection of 0.02 mL of 3.75% formalin into the left hind paw on the first and third days after administration. (**A**) Body weight, (**B**) intact paw thickness, (**C**) induced paw thickness, (**D**) differences in paw thickness and (**E**) induced paw volume were measured. The *p* value of the dexamethasone-treated group is presented as * *p* < 0.05 and ** *p* < 0.01 compared with the formalin control group. The *p* value of the PHT-treated group is presented † *p* < 0.05 and ‡ *p* < 0.01 compared with the formalin control group. The values are presented as the mean ± SD of nine experimental animals.

**Figure 5 molecules-25-03345-f005:**
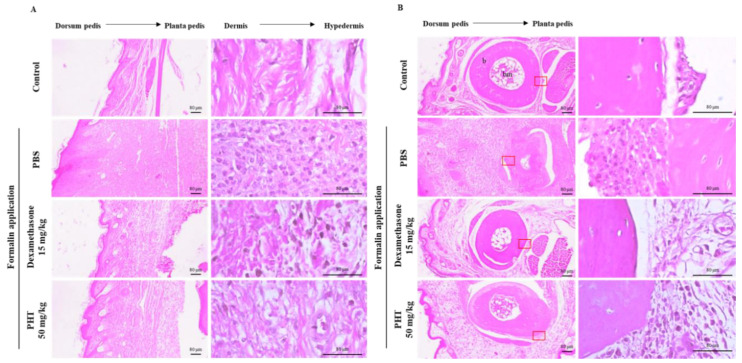
Histological analysis of induced paw (**A**) dorsum pedis skin and (**B**) digital skin. The paw was removed and fixed with 10% formalin. Then, samples were embedded in paraffin, sectioned with a microtome and stained with H&E. The data are representative of each group. b, bone; bm, bone marrow. Scale bars represent 80 μm as indicated.

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
