# Peer review of "Acyclic Triterpenoid Isolated from Alpinia katsumadai Alleviates Formalin-Induced Chronic Mouse Paw Inflammation by Inhibiting the Phosphorylation of ERK and NF-κB"

_molecules, 2020, doi:10.3390/molecules25153345_

Round 1

Reviewer 1 Report

The manuscript "Acyclic triterpenoid isolated from Alpinia katsumadai alleviate formalin-induced chronic mouse paw inflammation by inhibiting the  phosphorylation of ERK and NF-κB" describes a number of experiments in which the antiinflammatory activity of an isolated compound was investigated.

In the introduction more information can be supplied about this type of compound, e.g. have previous studies shown similar activities.

Some minor changes in the English will be necessary, e.g. in the title and line 63.

The figures are not very clear, because the letters are small and the plus and minus signs are not clear.

The experiments were performed with an isolated compound. Each isolation might provide a fraction with a different purity. Essential data of the isolated fraction should be given. NMR data and optical rotation of the isolated fraction and an image of the 1H NMR spectrum of the isolated compound.

The description of the isolation of the compound is not clear. In line 199 it is written that a fraction was subjected to column chromatography and compounds were obtained by TLC.

In line 254 is written "50 mg/kg PHT intraperitoneally and orally once a day for 10 days". Was the administration done intraperitoneally and orally at the same time??

Author Response

The manuscript "Acyclic triterpenoid isolated from Alpinia katsumadai alleviates formalin-induced chronic mouse paw inflammation by inhibiting the phosphorylation of ERK and NF-κB" describes a number of experiments in which the anti-inflammatory activity of an isolated compound was investigated.

Point 1: In the introduction more information can be supplied about this type of compound, e.g. have previous studies shown similar activities. 

Response 1: Thanks for your review. We added more information of acyclic triterpenoid in introduction part.

Point 2: Some minor changes in the English will be necessary, e.g. in the title and line 63.

Response 2: Thanks for your comment. We revised them.

Point 3: The figures are not very clear, because the letters are small and the plus and minus signs are not clear.

Response 3: Thanks for your review. We revised all figures following your comment.

Point 4: The experiments were performed with an isolated compound. Each isolation might provide a fraction with a different purity. Essential data of the isolated fraction should be given. NMR data and optical rotation of the isolated fraction and an image of the 1H NMR spectrum of the isolated compound.

Response 4: Thanks for your review. In our previous paper, we provided various information of suggested by the reviewer. “Acyclic Triterpenoids from Alpinia katsumadai Inhibit IL-6-Induced STAT3 Activation (Molecules. 2017 Oct; 22(10): 1611.)”. In addition, we added NMR data instead of image data in material and method part.

Point 5: The description of the isolation of the compound is not clear. In line 199 it is written that a fraction was subjected to column chromatography and compounds were obtained by TLC.

Response 5: Thanks for your review. We revised our manuscript following to your comment.

Point 6: In line 254 is written "50 mg/kg PHT intraperitoneally and orally once a day for 10 days". Was the administration done intraperitoneally and orally at the same time??

Response 6: Thanks for your review. We administrated saline and dexamethasone intraperitoneally and PHT orally. We revised manuscript as “the mice were administered saline and 15 mg/kg dexamethasone intraperitoneally and 50 mg/kg PHT orally once a day for 10 days”.

∴ We appreciate your review of our manuscript entitled “Acyclic triterpenoid isolated from Alpinia katsumadai alleviate formalin-induced chronic mouse paw inflammation by inhibiting the phosphorylation of ERK and NF-κB. (molecules-865769)”. We also believe that our manuscript has been improved due to your valuable comments and suggestions.

Reviewer 2 Report

This manuscript described interesting in vitro and in vivo anti-inflammatory properties with elucidation of the mechanisms of action of an acyclic triterpenoid from Alpinia katsumadai. Therefore, it deserves publication in Molecules after some minor revisions

1)

The Latin name  of the plant , the first time that is cited in the text has to be followed by the name of the taxonomist who identified the plant the first time

Furthermore the synonimies of the plant should be mentioned . Please check the “The Plant list 2013

--> Alpinia katsumadae Hayata is a synonym of Alpinia hainanensis K.Schum.

You should clearly describe the plant material and precise who has identified the seeds  and where is located the voucher specimen , number of voucher ?

2) The numbering of  the carbons of the PHT compound is missing

3) Paragraph 3.2. has to be revised. Although the obtention of the product is briefly described,  some important features are missing such as conditions of RP 18 separation (you need to precise the granulometry and pressure with which equipments . L 204 size of the column?

Furthermore, a sentence describing  the methodology  of characterization of the compound will be appreciated .

4) the name of the compound PHT  has to be changed:  “ …….6,10,14,18-tetracosatetraene  to be changed into ……..tetracosa-6,10,14,18-tetraene

5) p7/12 Fig 4: legend of subfigures days after administeration changed into "days after administration".  (check carefully all the subfigures A-F)

6) L 95, L 140, L 213: with the indicated concentrations of dexamethasone and PHT for 24 h. Could you write these concentrations?

7) L 236 dexamethasone and PHT for 1 h. could you indicate the concentrations?

8)L 277 could you write the phytochemical class of PHT?

9) the conclusion should be enlarged with a stronger discussion which will compare the activities reported by the authors in 2017 and the current studies.

Author Response

This manuscript described interesting in vitro and in vivo anti-inflammatory properties with elucidation of the mechanisms of action of an acyclic triterpenoid from Alpinia katsumadai. Therefore, it deserves publication in Molecules after some minor revisions

Point 1: The Latin name of the plant, the first time that is cited in the text has to be followed by the name of the taxonomist who identified the plant the first time. Furthermore the synonimies of the plant should be mentioned. Please check the “The Plant list 2013

Response 1: Thanks for your review. We revised it as “Alpinia katsumadai Hayata (Alpinia katsumadae Hayata) is a synonym of the Alpinia hainanensis K. Schum. and belong to the family Zingiberaceae”

Point 2: You should clearly describe the plant material and precise who has identified the seeds and where is located the voucher specimen, number of voucher. The numbering of  the carbons of the PHT compound is missing

Response 2: Thanks for your review. We revised it as “The authenticity of the plants was confirmed by Prof. Y. H. Kim, at the College of Pharmacy of Chungnam National University. A voucher specimen (PBC-386A) was deposited in the Korea Plant Extract Bank at the Korea Research Institute of Bioscience and Biotechnology”.

Point 3: Paragraph 3.2. has to be revised. Although the obtention of the product is briefly described, some important features are missing such as conditions of RP 18 separation (you need to precise the granulometry and pressure with which equipments. L 204 size of the column? Furthermore, a sentence describing the methodology of characterization of the compound will be appreciated.

Response 3: Thanks for your review. We revised manuscript of section 3.2.

Point 4: the name of the compound PHT  has to be changed:  “ …….6,10,14,18-tetracosatetraene  to be changed into ……..tetracosa-6,10,14,18-tetraene

Response 4: Thanks for your comment. We revised it.

Point 5: p7/12 Fig 4: legend of subfigures days after administeration changed into "days after administration".  (check carefully all the subfigures A-F)

Response 5: Thanks for your review. We revised Figure 4.

Point 6: L 95, L 140, L 213: with the indicated concentrations of dexamethasone and PHT for 24 h. Could you write these concentrations?

Response 6: Sorry for the confusion. The concentration was clearly stated in accordance with the reviewer's comment. Thank you.

Point 7: L 236 dexamethasone and PHT for 1 h. could you indicate the concentrations?

Response 7: Sorry for the confusion. The time and concentration were clearly stated in the manuscript according to the reviewer's comment. Thank you.

Point 8: L 277 could you write the phytochemical class of PHT?

Response 8: Thanks for your review. Based on the reviewer's comment, I have written a phytochemical class for PHT.

Point 9: the conclusion should be enlarged with a stronger discussion which will compare the activities reported by the authors in 2017 and the current studies

Response 9: Thanks for your review. We revised it.

Round 2

Reviewer 1 Report

The recommendations after the first review were accepted. The present version can be accepted for publication.